# Cascaded Stimulated Polariton Scattering in a Single-Pass KTP Crystal under Picosecond Pumping

Konstantin A. Vereshchagin [1], Alexey K. Vereshchagin [1], Vyacheslav B. Morozov [2,*] and Vladimir G. Tunkin [2]

[1] Prokhorov General Physics Institute of the Russian Academy of Sciences, Vavilov Str. 38, Moscow 119991, Russia; stinkapella@mail.ru (K.A.V.); iofan.fe@mail.ru (A.K.V.)
[2] Faculty of Physics, Lomonosov Moscow State University, Leninskie Gori 1, Moscow 119991, Russia; vladimirtunkin@mail.ru
[*] Correspondence: morozov@phys.msu.ru

**Abstract:** Cascaded stimulated polariton scattering (SPS) under powerful picosecond pumping of 532 nm wavelength was investigated in a single pass of a KTP ($KTiOPO_4$) crystal. Under ordinary polarization of the pump wave (relatively *XOZ*-plane) parametric scattering in the KTP crystal was observed. By rotation of the pump wave polarization (from ordinary to extraordinary), this scattering gradually transforms into polariton scattering. Polariton signal components (spots) with gaps between them were registered at external angles of ~2° between the neighboring spots. Five polariton signal components were detected above the pump beam, with a frequency difference between neighboring cascaded components of ~188 cm$^{-1}$ (5.64 THz). The wavelength of each next component increased regularly with the angle between this component and the pump wave, so this regular sequence of polariton signal components can be regarded as a cascaded SPS.

**Keywords:** stimulated polariton scattering; cascaded scattering; picosecond pumping; KTP crystal

## 1. Introduction

Polariton scattering has been studied from the first experimental work in 1970 until now. In the majority of these works, a pump beam of 1064 nm wavelength was used. Tunable THz polariton lasers based on polariton scattering in crystals and pumped by Q-switched lasers were realized in a number of papers [1–10]. Nonlinear crystals, LiNbO$_3$ [1–4], KTP [5–8], RTP [9] and KTA [10], were fixed on a rotating stage together with polariton cavity mirrors. This stage allowed the authors to tune the small angle $\theta$ between the polariton cavity axis and pump laser beam and in this way tune the signal (Stokes) and idler (THz) frequencies intermittently. The sufficiently short length of the THz polariton cavity (up to ~10 cm) allowed the authors to increase the intensity of the signal and THz waves during the nanosecond pumping pulses (7–30 ns). The polariton cavity is either inside the fundamental cavity [3–5,8–10] or outside it [1,2,6,7]. Intermittent tuning of the signal and THz waves in the process of the angle $\theta$ tuning was interpreted as the result of stimulated polariton scattering (SPS). To achieve relatively intense generation of the signal and consequently THz waves with a sufficiently narrow spectrum, a seeding radiation of an external tunable diode laser was injected into LiNbO$_3$ [2,11] and KTP [12] crystals in a version of THz sources without polariton cavity.

Picosecond pulses were also used to pump polariton amplifiers and lasers. In [13], a LiNbO$_3$ linear cavity of 1.68 m length was synchronously pumped by bunches of picosecond pulses of 7.5 ps duration. Ring cavities providing synchronous pumping to obtain SPS in LiNbO$_3$ crystal by picosecond pulses of 15 ps duration at 80 MHz repetition rate were realized in [14,15]. Femtosecond pulses were also used to generate THz radiation enhanced by phonon polaritons [16–18].

In several papers, cascaded SPS was realized [3,4,16]. In [3], up to four cascaded components were generated in a LiNbO$_3$ crystal which was placed in the polariton cavity.

Second and fourth components were generated in the main cavity and first and third components in the polariton cavity. In [4], first and second cascaded components were observed on each side of the pump beam. A parametric amplifier was realized in a KTP crystal under pumping by pulses of 520 ps duration and a wavelength of 1064.2 nm seeded by an external-cavity diode laser [19]. Four cascaded signal polariton components were observed. In [20], a two-pass excitation system based on a roof prism was used to increase the signal intensity initiated by a Q-switched pump beam in a KTP crystal. Two signal polariton components were observed on each side of the pump beam which propagated in the *XOY*-plane and were polarized parallel to the crystal Z-axis with a frequency difference between components of 192.3 cm$^{-1}$ (5.76 THz).

SPS was realized in KTP crystals in a single-pass geometry under quite powerful picosecond pumping: KTP crystals were pumped by 802 nm [21] and 1064 nm [22] picosecond pulses. In [21], two SPS spots were observed on each side of the pump beam: one generated by the pump beam and another by non-phasematched second harmonic. In [22], seven cascaded polariton components with a frequency difference of 5.6 THz between them and concentrated in two spots on each side of the pump beam and also in the central pump spot were generated. The pump beam in [21,22] was propagated in the XOY crystal plane and was polarized perpendicular to this plane and parallel to the crystal Z-axis.

A comprehensive review of the application of THz radiation in various fields of science and technology is given in [23,24].

In this paper, cascaded stimulated polariton scattering was investigated in a single-pass KTP crystal under pumping by powerful picosecond pulses of 75 ps duration and a wavelength of 532 nm. In contrast to [18,19], the pump beam propagated in the XOZ crystal plane and was polarized parallel to this plane. Non-phasematched second harmonic generation and therefore polariton scattering on its basis observed, for example, in [21] is excluded in our case due to strong KTP crystal absorption at wavelengths shorter than 350 nm. In [22], seven KTP crystal polariton components were measured in the pump beam spot and two spots above (or below) the pump beam. The aim of this work was to generate in a single-pass KTP crystal under powerful picosecond pumping a sequence of cascaded polariton components with a permanently increasing angle between each next polariton component and the pump beam.

## 2. Materials and Methods

In the present work, we have used the same *N*on-collinear *O*ptical *P*arametric *A*mplifier (NOPA) which we utilized in our earlier experiments [25,26]. The experimental setup is shown in Figure 1. The output of the picosecond laser (V "LOTIS TII", Minsk, Belarus) with 1–15 Hz repetition rate and 75 ps pulse duration at 1064 nm (fundamental frequency, FF) was frequency doubled, yielding pulses at 532 nm (second harmonic, SH) with a pulse duration of 63 ps. The laser worked at a 7 Hz repetition rate. SH radiation with pulse energy ~10 mJ pumped NOPA. In the present experiments, the pulse energy was increased by two times (up to ~20 mJ), so with beam width $w$~1.3 mm the intensity of the pump pulse reached values about 10 GW/cm$^2$.

*SH* beam was directed into the KTP crystal through a variable telescope formed by lenses L1 and L2 with focal lengths f = 300 mm and f = −100 mm, respectively. This telescope shapes *SH* beam to control the diameter and the divergence of the pump beam inside the KTP crystal. A "beam-stopper" (d~6 mm), blocking the powerful pump beam, could be placed behind the crystal.

The KTP crystal (6 mm × 6 mm × 10 mm) was cut for non-collinear parametric phasematching in the *XOZ*-plane (horizontal) of waves with wavelengths $\lambda_p$ = 532 nm, $\lambda_s$ = 684 nm, and $\lambda_i$ = 2382 nm, with the angle between the crystal Z-axis and the normal to its surface $\theta_p$~62°. In the case of parametric interaction («*NOPA*-mode»), this was a *type II* interaction, where the pump ($\lambda_p$) and idler ($\lambda_i$) waves were polarized in the vertical plane and the signal wave ($\lambda_s$) was polarized horizontally. Since the output second harmonic radiation from the picosecond laser was polarized horizontally, a $\lambda/2$ phase plate (**PP**) was

used to rotate the pump wave polarization at angles from 0° to 90° (from horizontal to vertical one).

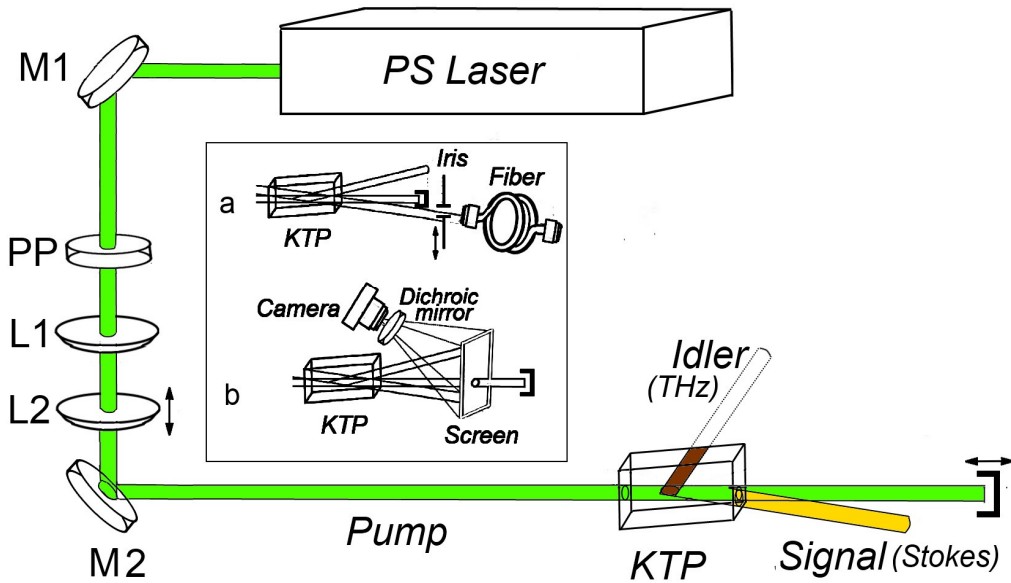

**Figure 1.** Outline of the experimental setup. M1, M2 are dielectric mirrors, Pump is second harmonic beam, PP is λ/2 phase plate, L1 and L2 are spherical lenses forming a variable telescope. On the inset: layout of the equipment for data collection; (a) wavelength measurements, (b) taking pictures and videos.

Emission from the NOPA forms the complex spatial pattern. The screen was located behind the KTP crystal at the distance of ~195 mm with its surface perpendicular to the pump beam; there was a hole in the screen (~6 mm) for passing through of the direct pump beam. We took pictures of the multicolor image on the screen, analyzed the polarization of each pattern of the image, and measured its spectra.

The image of the visible scattered light distribution on the screen was photographed by a conventional camera through either the dichroic mirror or color orange filter *OS*-12 installed in front of the camera.

The dichroic mirror worked like a bandpass filter: it reflected the pump radiation and to some extent transmitted radiation from other spectral ranges. Therefore, the color rendition of pictures on the screen could be a little distorted.

Light polarization of all the image patterns was analyzed using a linear film polarizer placed behind the "stopper". In all the cases the polarization was linear, being either horizontal or rotated at small angle from horizontal. When the spectra were recorded, the screen was removed and the radiation from the KTP crystal output was transmitted to the spectrograph through optical fiber. For laser-like beams, all the area of the light spot was focused by the micro-objective onto the entrance face of the multimode optical fiber (diameter ~600 μm). When recording the spectra of individual parts of a multi-color picture, distributed over the space of the entrance surface of the micro-objective, a small diaphragm was set in the corresponding place of the picture, while the micro-objective focusing light at the fiber input was located just beyond the diaphragm. Such approach was valid only for the visible and intense enough to be seen light; it was accompanied by a lengthy adjustment of the fiber input when the diaphragm position at the picture was changed. The fiber exit end was projected by means of an objective onto the spectrograph entrance slit. The spectra were analyzed with the help of monochromator–spectrograph (JSC "LOMO", Saint Petersburg, Russia) combined with the CCD detector (S7031-1007S, Hamamatsu). The spectrograph entrance slit typically was 200 μm wide.

### 3. Results

#### 3.1. Outside the Crystal

The light pattern on the screen behind the crystal of a given configuration (orientation of the crystallographic axes with respect to the crystal faces, incidence angle of the pump beam on the input face) depends fundamentally on the polarization of the pump wave (Figure 2). Since the crystal was designed to implement a non-collinear parametric amplifier (NOPA) in the red region of the spectrum [25], the working plane (*XOZ*) was horizontal and the polarization of the pump wave had to be vertical (*ordinary*). Light patterns of complex configurations were observed in the plane of the screen due to the processes of *down-* and *up-conversion* [26] with subsequent amplification of these light waves in the NOPA (Figure 2a–d). The spectral range of radiation from these processes included both visible light (*VIZ*) and near-*IR* radiation (*NIR*).

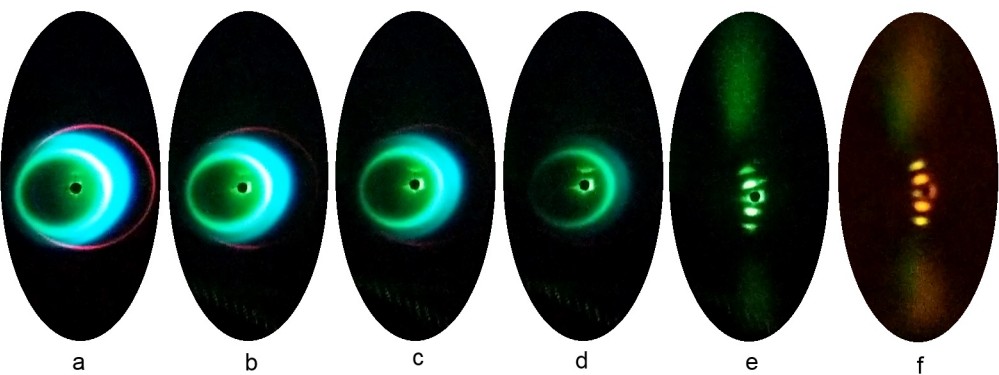

**Figure 2.** (**a**–**e**): Evolution of the light pattern on the screen as pump wave polarization was changed from vertical (*ordinary*) to horizontal (*extraordinary*). Parametric superfluorescence produces the thin red ring (**a**,**b**) and round *IR* spot inside this ring. Blue, cyan, and green rings are the result of up-conversion processes. (**e**–**f**): Image obtained when photographing through a dichroic mirror, suppressing pump radiation at a wavelength of 532 nm € and using an *OS*-12 orange light filter (**f**). In the last case, the color rendition is disturbed due to the saturation of the camera matrix. The angle of incidence of the pump beam on the crystal is 22°. In the center of the screen there is a hole for passing through the intense pump beam.

When the polarization of the pump wave was changed from vertical (*ordinary*) to horizontal (*extraordinary*), the light pattern on the screen in the visible range (*VIZ*) was radically changed (Figure 2e,f).

With horizontal polarization of the pump wave (*extraordinary*), the most intense parts of the light pattern are elongated spots in the "beans" form, the centers of which are located on some arc passing through the center of the pump beam spot (Figure 3). Two spots (*beans*) are stably reproduced in each laser flash: above ("positive" *bean* number) and below ("negative" *bean* number) the pump beam. As the energy in the pump pulse $E_{puls}$ increases, initially ($E_{puls} \sim 1/2(E_{puls})_{max}$) the first spots pair ($\pm 1$ *beans*) appears, and then ($E_{puls} \sim 3/4(E_{puls})_{max}$) the *beans* of the second spots pair arise ($\pm 2$ *beans*). At the maximum energy of the pump pulse ($E_{puls} = (E_{puls})_{max}$), the *beans* of the higher orders appear from time to time. We explain the instability of their appearance by fluctuations in the duration of the laser pulse and its energy. With a sporadic single-shot increase in the laser pulse energy and/or a decrease in its duration, the peak intensity of the pump wave increases. In turn, it leads to a significant increase in the efficiency of the nonlinear optical processes and parametric amplification. Attempts to raise the pump wave intensity either by increasing the pulse energy or by focusing the pump beam gave a short-term positive result, which was very soon followed by an optical breakdown of the crystal or, in a more "soft" mode, clouding it due to the formation of "gray tracks". Thus, for the study of *beans* of the higher orders, we were forced to use video recording to capture images of the light pattern and to

implement the signal accumulation modes when measuring wavelengths in *beans* of orders higher than the second.

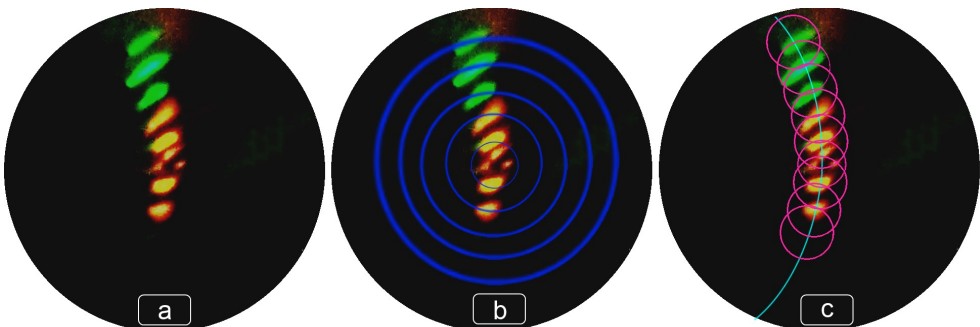

**Figure 3.** Locations of the light spots on the screen (**a**). The equidistance of the *beans* locations is illustrated using a "guide for eyes" in the form of concentric rings centered on the pump beam spot (**b**). Every *bean* has two equidistantly remote (*up* and *down*) satellites with the arc-trajectory of their centers in the screen plane (**c**); all the circles and the arc are the "guides for eyes". The hole in the screen with a diameter of 6 mm is destined for the pump beam to pass through it. There are five components above the hole and two below it since the pump beam has been specially adjusted to cross the crystal at a slight angle to the XOZ-plane.

The coordinates of the light spots shown in Figure 3, in a spherical coordinate system linked to the center of the pump spot on the screen, are given in Table 1. From Figure 3 and Table 1 it seems that the beans are located almost equidistantly, and the radii and polar angles are approximately arithmetic progressions with steps $\Delta R \sim 12$ mm and $\Delta\theta \sim 3.5°$, respectively. The distances of the observed light spots from the pump beam spot along with the trajectory of their centers in the screen plane *outside* the crystal determine the polar and azimuthal angles of the wavevectors of these waves in the main crystallographic axes *inside* the crystal. The wavelengths of radiation corresponding to the light pattern have been measured and are also given in Table 1.

**Table 1.** The distance (R) of the observed light spots from the pump beam, polar, and azimuthal angles of the wavevectors of these waves and corresponding wavelengths.

| *Bean* Number | R, mm | Polar Angle, Grad | Azimuth Angle, Grad | Wavelength, nm |
|---|---|---|---|---|
| 1 | $12.9 \pm 0.3$ | $3.76 \pm 0.10$ | $-84.3 \pm 1.0$ | $537.37 \pm 0.02$ |
| 2 | $24.8 \pm 0.2$ | $7.64 \pm 0.04$ | $-84.4 \pm 1.0$ | $542.85 \pm 0.05$ |
| 3 | $36.7 \pm 0.2$ | $11.02 \pm 0.04$ | $-81.9 \pm 1.0$ | $548.45 \pm 0.05$ |
| 4 | $50.6 \pm 0.2$ | $14.87 \pm 0.04$ | $-78.9 \pm 1.0$ | $553.90 \pm 0.05$ |
| 5 | $63.4 \pm 0.2$ | $18.31 \pm 0.04$ | $-76.7 \pm 1.0$ | - |
| −1 | $11.2 \pm 0.3$ | $3.28 \pm 0.10$ | $79.5 \pm 1.0$ | $537.37 \pm 0.02$ |
| −2 | $24.5 \pm 0.2$ | $7.16 \pm 0.04$ | $82.2 \pm 1.0$ | $542.85 \pm 0.05$ |

Figure 4 shows the spectra measured in *beans*. The relative intensities of the spectral components in each spectrum depend strongly on the angle of their entry into the fiber transmitting light to the spectrograph. This angle can be adjusted somewhat by tilting the fiber in the desired direction for better detection of radiation in the corresponding *bean*. It is easy to see that the line (third *bean*) at a wavelength of 545.45 nm (Figure 4c) is very strong but practically not visible in Figure 4d, where the fourth spectral component (fourth *bean*) is detected at a wavelength of 553.9 nm, although they correspond to the neighboring *beans*. In the video mode, we managed to register even five *beans* above the hole, but the frames with the appearance of the fifth *bean* were so rare and the corresponding, even accumulated, signal was so small that we could not adjust the angle of radiation input of the fifth *bean* into a fiber to measure its spectrum.

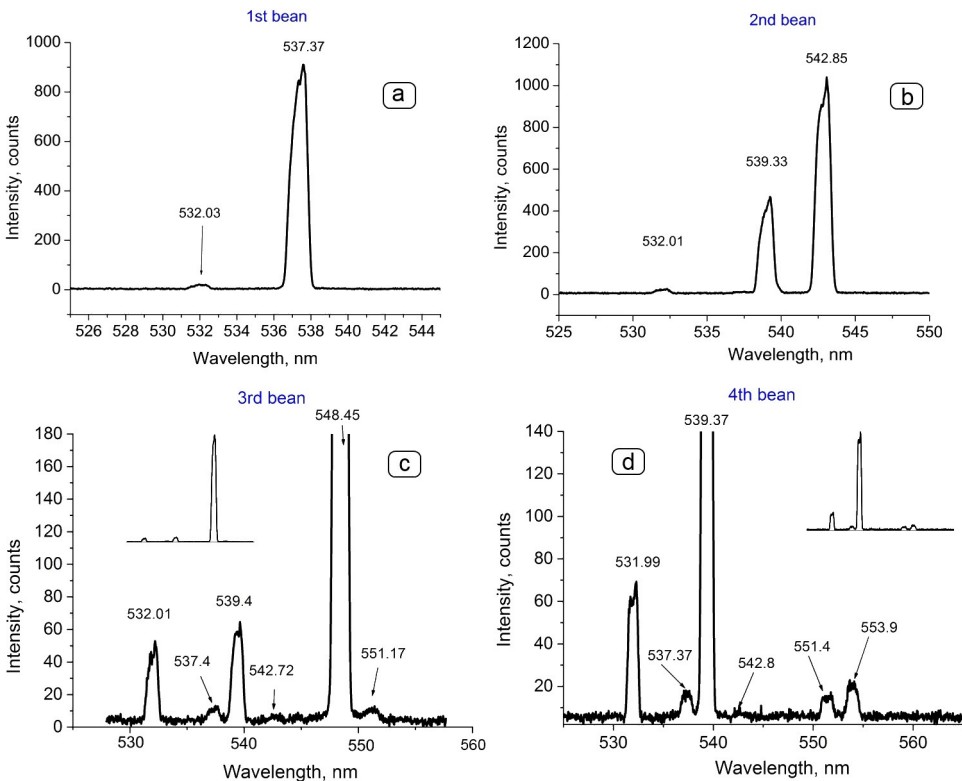

**Figure 4.** The cascaded SPS spectra measured in KTP crystal in different polariton spots (*beans*) above the hole: (**a**)—1st *bean*, (**b**)—2nd *bean*, (**c**)—3rd *bean*, and (**d**)—4th *bean*. The spectra in (**c**,**d**) are zoomed to demonstrate additional weak lines in the recorded spectra. The insets in (**c**,**d**) show the full-scale spectra.

The following sequence of wavelengths was measured: first *bean*—537.37 nm, second—542.85 nm, third—548.45 nm, fourth—553.9 nm. The differences between the pump frequency and the first signal component, between the first and the second components, between the second and the third components, between the third and the fourth components were 187.84 cm$^{-1}$, 187.85 cm$^{-1}$, 188.09 cm$^{-1}$, and 179.4 cm$^{-1}$, respectively. Such frequency difference points out a process that links all light beams under consideration. The candidate for this process is polariton scattering since the data available from the literature [17–19] give also a value of ~188 cm$^{-1}$ for the polariton frequency in KTP. It seems that the sequence of spectral components observed can be represented as a cascaded **SPS** process: the pump wave generates the first polariton component, the corresponding Stokes wave generates the second polariton component, etc., with an increasing angle between the wavevector of each subsequent Stokes component and the pump wavevector.

### 3.2. Inside the Crystal

The characteristics of a polariton wave in a crystal can be obtained from the features of the pump and Stokes waves, which are observed and recorded (wavelength and direction of propagation) *outside* the crystal. Since the properties of all waves of the processes under consideration depend on the direction of their propagation, the initial task is to find the directions of wave propagation *inside* the crystal. The external angles (*outside* the crystal) between the observed light beams can be found from the relative position of the corresponding light spots on the screen (See Figure 3). The calculation of the internal angles (*inside* the crystal) is complicated by the fact that the refractive index on the output face of the crystal of all observed rays depends on the angle of refraction.

Thus, in addition to directly measuring the wavelengths of the Stokes components of SPS processes, we need to determine the refractive indices for the pump wave and the Stokes waves of the corresponding cascade processes.

It should be remembered that the crystal is cut in such a way that the working principal plane (*XOZ*) containing both optical axes is located horizontally and the *Z*-axis coincides with the bisector of the angle between the axes, and the angle $\theta_0$ between the *Z*-axis and the normal to the surface of the input/output face of the crystal is 62°. A biaxial crystal with the propagation of rays in one of the principal planes of the crystal is similar to a uniaxial one [27], and since the polarization of the pump wave is horizontal, the pump wave in the *XOZ*-plane is *extraordinary*. For Stokes waves, the refractive index also depends on the direction of propagation in the crystal. However, the case here is even more complicated than with the refractive index of the pump wave: as can be seen from Figure 3, the wavevector of any Stokes wave quits the horizontal plane *XOZ*.

### 3.2.1. Rays in the Principal Plane XOZ

At a given angle $\gamma_{ext}$ of the pump beam incidence on the input face of the crystal, the refraction angle $\gamma_{int}$ depends (through the refractive index of the extraordinary ray) on the direction of the ray with respect to the *Z*-axis and, therefore, depends on the incidence angle (See Figure 5). Indeed,

$$n_e(\theta,\lambda) = \frac{n_x(\lambda)\cdot n_z(\lambda)}{\sqrt{n_x^2(\lambda)\cdot\sin^2(\theta) + n_z^2(\lambda)\cdot\cos^2(\theta)}}, n_o(\theta,\lambda) = n_o(\lambda) = n_y(\lambda) \tag{1}$$

and, according to the Snell's law,

$$\theta = \theta_0 + \gamma_{int} = \theta_0 + \arcsin\left(\frac{\sin(\gamma_{ext})}{n_e(\theta,\lambda_{pump})}\right) \tag{2}$$

Combining (1) and (2) and denoting the refractive index of the pump wave in a given direction as $n_{pump} = n_e(\theta,\lambda_{pump})$, we obtain a nonlinear equation with respect to $n_{pump}$:

$$n_{pump} = \frac{n_x(\lambda_{pump})\cdot n_z(\lambda_{pump})}{\sqrt{n_x^2(\lambda_{pump})\cdot\sin^2\left(\theta_0 + \arcsin\left(\frac{\sin\gamma_{ext}}{n_{pump}}\right)\right) + n_z^2(\lambda_{pump})\cdot\cos^2\left(\theta_0 + \arcsin\left(\frac{\sin\gamma_{ext}}{n_{pump}}\right)\right)}} \tag{3}$$

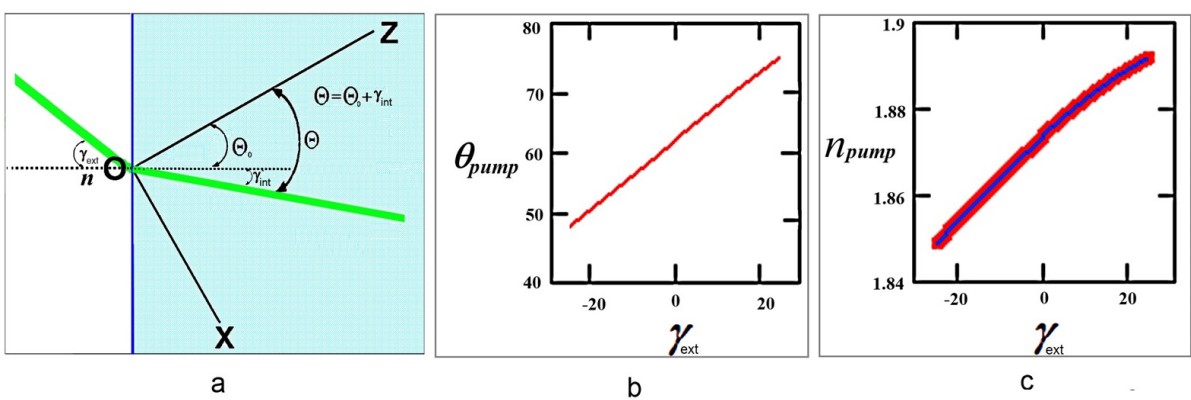

a           b           c

**Figure 5.** (**a**): To the refraction of light at the crystal face. (**b,c**): Polar angle $\theta_{pump}$ and refractive index $n_{pump}$ of the extraordinary pump beam ($\lambda_{pump}$ = 532 nm) in the main plane *XOZ* of the KTP crystal as the function of the incident angle $\gamma_{ext}$. (**c**): The bold red curve is solution of Equation (3), the thin blue curve is $n_{pump} = F_{pump}(\gamma_{ext})$. The angle $\gamma_{ext}$ shown in (**a**) corresponds to the positive values of $\gamma_{ext}$, for which $\theta > \theta_0$.

Here $n_x$, $n_y$, and $n_z$ are the principal refractive indices of the crystal at a given wavelength [28]. Equation (3) is solved numerically for any given incidence angle $\gamma_{ext}$. By interpolating (*spline procedure*) solutions of Equation (3) for a set of incidence angles from $-25°$ to $+25°$, the formula $n_{pump} = F_{pump}(\gamma_{ext})$ was obtained to calculate the refractive index of the pump wave for any incidence angle $\gamma_{ext}$ in the specified range of angles.

### 3.2.2. Rays in Any Direction

To find the refractive index $n(\hat{s})$ of the ray in a given direction $\hat{s} = (s_x, s_y, s_z)$, we use the indicatrix equation given by Fresnel's equation of wave normals, expressed in terms of the crystal principal crystallographic axes [29]:

$$\frac{s_x^2}{\frac{1}{n^2(\hat{s})} - \frac{1}{n_x^2}} + \frac{s_y^2}{\frac{1}{n^2(\hat{s})} - \frac{1}{n_y^2}} + \frac{s_z^2}{\frac{1}{n^2(\hat{s})} - \frac{1}{n_z^2}} = 0 \tag{4}$$

For a biaxial crystal, $n_x < n_y < n_z$, while for a uniaxial crystal, $n_x = n_y = n_o$ (*ordinary*) and $n_z = n_e$ (*extraordinary*). Following the work [30], Equation (5) can be rewritten as

$$x^2 - \left[ s_x^2 \cdot \left( \frac{1}{n_y^2} + \frac{1}{n_z^2} \right) + s_y^2 \cdot \left( \frac{1}{n_x^2} + \frac{1}{n_z^2} \right) + s_z^2 \cdot \left( \frac{1}{n_x^2} + \frac{1}{n_y^2} \right) \right] \cdot x + \left[ \frac{s_x^2}{n_y^2 \cdot n_z^2} + \frac{s_y^2}{n_x^2 \cdot n_z^2} + \frac{s_z^2}{n_x^2 \cdot n_y^2} \right] = 0 \tag{5}$$

with $x = \frac{1}{n^2(\hat{s})}$. Solving (5) with respect to the variable $x$, we obtain one solution for each possible polarization (*fast* or *slow* waves instead of *ordinary* and *extraordinary* ones in a uniaxial crystal):

$$\begin{aligned} n_{fast} &= \sqrt{\frac{2}{B + \sqrt{B^2 - 4C}}}, \\ n_{slow} &= \sqrt{\frac{2}{B - \sqrt{B^2 - 4C}}} \end{aligned} \tag{6}$$

where

$$\begin{aligned} B &= s_x^2 \cdot \left( \frac{1}{n_y^2} + \frac{1}{n_z^2} \right) + s_y^2 \cdot \left( \frac{1}{n_x^2} + \frac{1}{n_z^2} \right) + s_z^2 \cdot \left( \frac{1}{n_x^2} + \frac{1}{n_y^2} \right) \\ C &= \frac{s_x^2}{n_y^2 \cdot n_z^2} + \frac{s_y^2}{n_x^2 \cdot n_z^2} + \frac{s_z^2}{n_x^2 \cdot n_y^2} \end{aligned} \tag{7}$$

The direction of the pump beam is set in the laboratory coordinate system. Since the crystallographic axes of the crystal are inconvenient for calculating the resulting output signal, we express the wavevectors of any wave in the laboratory frame linked with the pump beam. In the laboratory frame, the unit vectors of the pump and any other wave are given as

$$\begin{aligned} \hat{s}_{pump} &= s_{pump}(\theta_{pump}, \varphi_{pump}) \\ \hat{s}_s &= s_s(\theta_s, \varphi_s) \end{aligned} \tag{8}$$

where $s_i$ (*i = pump, s*) are the spherical coordinates of the unit vectors for waves in the given propagation direction $\hat{s}_i$. Here, $\theta_{pump}$ is the polar angle between $\hat{s}_{pump}$ and the $\hat{z}$ axis, and $\varphi_{pump}$ is the azimuthal angle (around $\hat{z}$) from the $\hat{x}$ axis to $\hat{s}_{pump}$ in the *XOY*-plane. For rays in an arbitrary direction, $\hat{s}_s$, angle $\theta_s$ specified relative to $\hat{s}_{pump}$ and the azimuthal angle $\varphi_s$ refers to rotations in the plane normal to $\hat{s}_{pump}$.

To calculate the refractive index of the *slow* signal wave, we use the following algorithm. First, we translate the coordinates of the propagation vector $\hat{s}_s$, given in the lab frame, into the coordinate system linked with the crystallographic axes of the crystal (*X, Y, Z*) and find the roots of Equation (4) using (6) and (7). At this stage, we do not know the true angles $\theta_s$, $\varphi_s$, so we build the surface of the refractive indices for plenty of directions *inside* the crystal (see Figure 6). For each of the directions, we find the angle of incidence onto output face and, using the corresponding value of the refractive index, determine the direction of the beam at the outlet from the crystal. Further, we collate the calculated direction of the beam *outside* the crystal with the positions of the light spots on the screen (Figure 3, Table 1). Finally, we select from the set of considered directions of the ray propagation *inside* the crystal a single pair of spherical coordinates ($\theta_s$, $\varphi_s$), matching the appropriate light beam direction. For direction found, we calculate the unique value of the refractive index $n_{slow}$ (single point at the surface in Figure 6a).

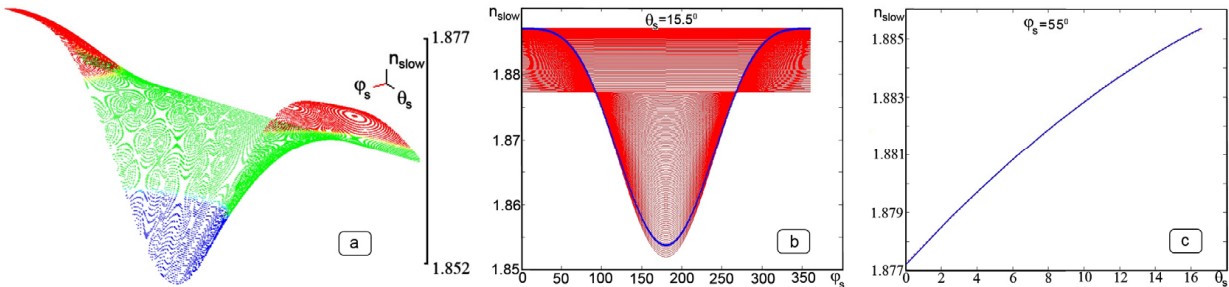

**Figure 6.** The surface of the refractive indices $n_{slow}$ (fragment of the indicatrix (Equation (4))) (**a**) and the cross sections of this surface (**b**,**c**) for the Stokes wave of the SPS first cascade with $\lambda_S$ = 537.4 nm; thin blue lines correspond to cross sections at $\theta_S$ = 15.5° (**b**) and $\phi_S$ = 55° (**c**).

Refractive indices $n_{slow}$ and unit vectors (propagation directions) of the Stokes waves for the observed SPS cascades at the angle of incidence of the pump beam on the input face of the crystal 22° are given in Table 2. Polar and azimuthal angles correspond to the spherical coordinates ($\theta$, $\phi$) determined in the frame linked with the crystallographic axes ($X$, $Y$, $Z$) of the crystal.

**Table 2.** Refractive indices $n_{slow}$ of the Stokes waves in the certain propagation directions. Spherical coordinates ($\theta$, $\varphi$) are given in the frame linked with the crystallographic axes ($X$, $Y$, $Z$) of the crystal.

| *Bean* Number | $n_{slow}$ | $\theta$, Grad | $\varphi$, Grad | Wavelength, nm |
|---|---|---|---|---|
| Pump | 1.890 | 74.430 | 0 | 532.00 ± 0.02 |
| 1 | 1.877 | 73.245 | 2.080 | 537.37 ± 0.02 |
| 2 | 1.875 | 72.993 | 3.974 | 542.85 ± 0.05 |
| 3 | 1.873 | 72.211 | 5.609 | 548.45 ± 0.05 |
| 4 | 1.871 | 71.790 | 7.752 | 553.90 ± 0.05 |
| −1 | 1.877 | 73.216 | −1.724 | 537.37 ± 0.02 |
| −2 | 1.875 | 72.993 | −3.974 | 542.85 ± 0.05 |

### 3.2.3. Characters of the Polariton Waves

Energy conservation law and phasematching conditions should be fulfilled for the polariton scattering:

$$\omega_{pump} = \omega_{St} + \omega_{pol} \tag{9}$$

$$k_{pump} = k_{St} + k_{pol} \tag{10}$$

where $\omega_{pump}$, $\omega_{St}$, $\omega_{pol}$ are frequencies and $k_{pump}$, $k_{St}$, $k_{pol}$ are wavevectors of the pump, Stokes and polariton waves, respectively. The phasematching triangle for polariton scattering is shown in Figure 7. The next cascaded SPS Stokes wave serves as the pump and so on.

By the cosine theorem for the phasematching triangle (10) with $k_{pump}$, $k_{St}$, $k_{pol}$, we obtain

$$k_{pol} = \frac{n_{pol}}{\lambda_{pol}} = \sqrt{\frac{n_{pump}^2}{\lambda_{pump}^2} + \frac{n_{St}^2}{\lambda_{St}^2} - 2\frac{n_{pump}n_{St}}{\lambda_{pump}\lambda_{St}}cos\alpha} \tag{11}$$

To calculate $n_{pol}$ from (11) we need to find only the phasematching angle $\alpha$ since the wavelengths and refractive indices for the pump and Stokes beams are already known (Table 2), and $\omega_{pol}$ can be found from (9). The unit vectors (8) for all beams under consideration have been determined when calculating the refractive indices. Thus, we find the sequence of $\alpha$-angles for the SPS cascades from the scalar products of the corresponding unit vectors:

$$\cos\alpha_1 = \left(\hat{s}_{pump}, \hat{s}_{Stokes_1}\right), \ 1^{st} \ stage$$
$$\cos\alpha_{i+1} = \left(\hat{s}_{Stokes_i}, \hat{s}_{Stokes1_{i+1}}\right), \ (i = 1, 2, 3), \ (i+1)^{th} \ stage \tag{12}$$

With known wavevectors values, another phasematching angle, $\beta$, can be calculated from phasematching triangle by the formula:

$$\cos \beta_1 = \frac{k_{pump} - k_{Stokes1} \cos \alpha_1}{k_{pol_1}}, \ 1^{st} \ stage$$
$$\cos \beta_{i+1} = \frac{k_{Stokes\,i} - k_{Stokes\,i+1} \cos \alpha_{i+1}}{k_{pol_{i+1}}}, \ (i = 1, 2, 3), \ (i+1)^{th} \ stage \tag{13}$$

Refractive indices $n_{pol}$ and phasematching angles ($\alpha$ and $\beta$) of the polariton waves linked with Stokes waves for the observed SPS cascades are given in Table 3.

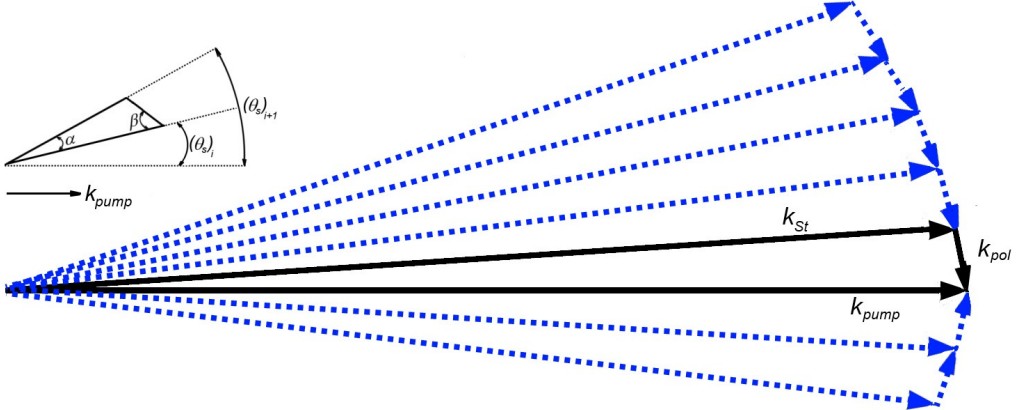

**Figure 7.** Phasematching triangles for different supposed cascaded SPS processes (5 above and 2 below the pump beam). $\mathbf{k_{pump}}$, $\mathbf{k_{St}}$, and $\mathbf{k_{pol}}$ are pump, signal, and polariton wavevectors, respectively. $\alpha$ and $\beta$ are phasematching angles, $(\theta_s)_i$ and $(\theta_s)_{i+1}$ are the polar angles of the serial Stokes (signal) beams in the lab frame. According to Figure 3, signal wavevectors are not lying in the same plane.

**Table 3.** Refractive indices $n_{pol}$ and phasematching angles ($\alpha$ and $\beta$) of the polariton waves linked with Stokes waves in the certain propagation directions. The angle of incidence of the pump beam on the input face of the crystal is $22°$.

| *Bean* * Number | $n_{pol}$ | $\alpha$ **, Grad | $\beta$ ***, Grad | Frequency, cm$^{-1}$/THz |
|---|---|---|---|---|
| 1 | 7.281 | 2.0 | 62.9 | $187.84 \pm 0.70 / 5.64 \pm 0.02$ |
| 2 | 6.265 | 1.8 | 69.8 | $187.86 \pm 1.00 / 5.64 \pm 0.03$ |
| 3 | 5.949 | 1.7 | 68.3 | $188.09 \pm 0.03 / 5.640 \pm 0.001$ |
| 4 | 7.181 | 2.0 | 72.2 | $179.40 \pm 0.03 / 5.380 \pm 0.001$ |
| $-1$ | 6.321 | 1.7 | 58.7 | $187.84 \pm 0.70 / 5.64 \pm 0.02$ |
| $-2$ | 7.283 | 2.1 | 72.5 | $187.86 \pm 1.00 / 5.64 \pm 0.03$ |

* polariton wave, related to corresponding Stokes wave; ** the angle between pump and signal wavevectors in the phasematching triangle; *** the angle between pump and idler wavevectors in the phasematching triangle.

## 4. Discussion

Now, we can explain the metamorphosis that occurs with the light pattern on the screen behind the crystal when the polarization of the pump wave is rotated (Figure 2). As indicated in [19], polaritons being coupled with vibrational modes of A$_1$ symmetry (similar to the phonon mode at a frequency of 5.7 THz) have a dipole moment oriented along the Z-axis. In the *NOPA*-configuration (Figure 2a), with vertical polarization of the pump wave, the projection of the electric field vector $\mathbf{E_{pump}}$ on the Z-axis is equal to zero and vibrational modes of symmetry A$_1$ are not excited; as the plane of polarization rotates, the projection of the electric field vector $\mathbf{E_{pump}}$ onto the Z-axis increases.

An oscillating dipole emits electromagnetic waves; the radiation of the dipole does not have spherical symmetry: it is maximal in the direction perpendicular to the axis of the dipole and is equal to zero along its axis. The radiation pattern of the dipole is symmetrical with respect to the direction of its axis, so in a *3D* image it looks like a bagel with an infinitesimal hole in the middle (Figure 8).

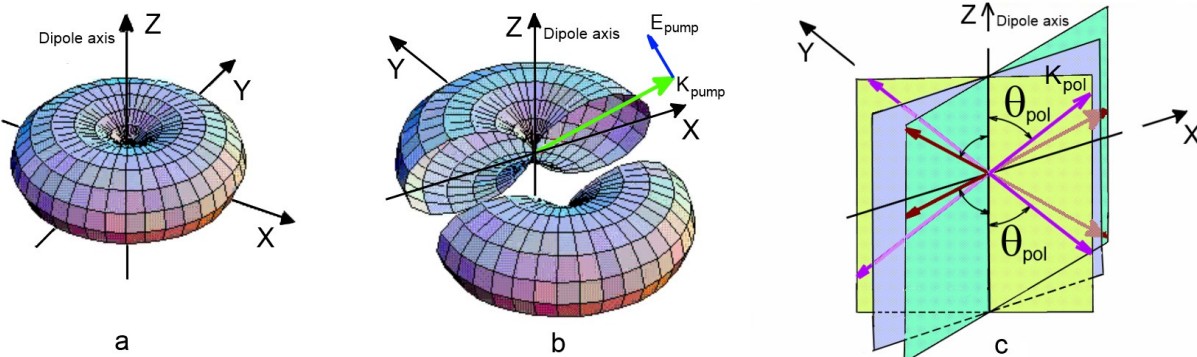

**Figure 8.** Radiation pattern of the dipole (**a**); pump wavevector $k_{pump}$ and the projection of the electric field vector $E_{pump}$ onto the Z-axis when both vectors $k_{pump}$ and $E_{pump}$ belong to the *XOZ*-plane (**b**); symmetry of polariton wavevectors $k_{pol}$ relatively Z-axis (**c**).

Due to birefringence in some directions in the crystal, the phasematching conditions will be fulfilled: the wave vectors of the pump, Stokes, and polariton waves form a closed triangle. Namely, in these directions the stimulated polariton scattering process will develop. With this, the enhanced spontaneous polariton scattering forms a light beam that gives a spot in the form of a *bean* on the screen.

The closer the pump wavevector $k_{pump}$ to the X-axis (both vectors $k_{pump}$ and $E_{pump}$ belong to the *XOZ*-plane) the larger the Z-component of the pump electric field vector $E_{pump}$ (Figure 8b). For this reason, the largest number of *beans* in the light pattern on the screen was observed at large angles of pump incidence on the input face of the crystal ($\gamma \sim 21°\ldots 25°$) when the angle of the pump wavevector $k_{pump}$ with the Z-axis is greatest (at $\gamma > 25°$ the pump beam does not pass through the crystal aperture); on the contrary, when the sign of the pump incidence angle changes ($\gamma \sim -21°$), the Z-component of the pump electric field vector $E_{pump}$ decreases (see Figures 5 and 8b), the efficiency of the cascade SPS decreases, and *beans* of orders higher than the second are not observed.

The role of the Z-axis component of the pump electric field in the cascaded polariton components generation is shown in [19]. The crystal in this work was cut for frequency doubling at 1064 nm using type-II phasematching, i.e., at the angles $\theta \sim 90°$ and $\phi \sim 21.3°$ to the principal axes. The pump beam propagated in the *XOY*-plane (horizontal) and polarization of this beam varied from the vertical to horizontal direction. In the first case, i.e., when the pump electric field was parallel to the Z-axis, seven polariton components at a $\sim 188 \text{ cm}^{-1}$ interval between them were observed. The number of polariton components diminished with an increase in the angle between the Z-axis and pump electric field, and, at the same time, the second harmonic intensity generated in the KTP crystal according to the type-II interaction (*oe-o*) increased.

Since the dipole axis of the vibrational mode associated with polaritons is oriented along the Z-axis, the wavevector of the polariton wave tends to be localized outside the *XOZ*-plane. Accordingly, the Stokes rays leave the *XOZ*-plane (Figure 3), and for each subsequent SPS cascade, both the polar angle θ and the azimuthal angle φ are changed (see Table 2). These changes in the direction of propagation of Stokes beams cause a curvature of the localization trajectory of the corresponding light spots on the screen (Figure 3c).

Additionally, as follows from the symmetry of the radiation pattern of the dipole (Figure 8), in its section by any plane parallel to the Z-axis there are two directions of polariton propagation with the same projections of the wavevector $k_{pol}$ on the Z-axis and with the same polarization of the electric field (Figure 8c). This circumstance causes the appearance of Stokes beams above and below (symmetrically) the pump beam, and, hence, the *beans* on the screen with «positive» and «negative» *bean numbers* (Figure 3). With respect to the SPS cascades, this means that in each subsequent cascade, where the Stokes wave of the previous cascade plays the role of the pump beam, beams with the wavelengths of the Stokes components appear symmetrically above and below the Stokes wave of

the preceding cascade. This circumstance is displayed in Figure 3c by means of circles with centers placed on the trajectory of localization of light spots on the screen. Thus, in the vicinity of each *bean* there are directions where it is possible to register the radiation of the Stokes components of subsequent cascades. As indicated in the commentary of Figure 4, there are certain difficulties with adjusting the input of radiation into the fiber when registering Stokes components of cascades with numbers greater than the second. Therefore, we do not give examples of such spectra; however, we note that a similar combination of the directions of higher-order Stokes waves was observed in [19].

As can be seen from Table 3, the angle $\alpha$ in the triangle of wavevectors for each cascade (Figure 7) remains approximately equal to 2°, which explains the approximate equidistance of the light spots in Figure 3b. The magnitudes of the angle $\beta \sim 60$–70°, along with high values of the refractive index of the polariton wave $n_{pol} \sim 6$–7, indicate a complete internal reflection of the polariton waves from any of the faces for a crystal having the shape of a parallelepiped. Thus, in the geometry of our experiment, we cannot observe terahertz radiation at the outlet of the crystal.

Obviously, the frequencies of the polariton waves of the first three cascades are approximately the same and differ markedly from the frequency of the polariton in the fourth cascade (Table 3). One possible explanation is that we mistook the radiation with a wavelength of 553.9 nm for the Stokes component of the fourth cascade; indeed, many processes of nonlinear interaction of light waves can be observed at the output of a nonlinear crystal with intense picosecond pumping. However, the totality of the experimental data testifies in favor of another hypothesis.

In a cascaded *SPS* process, the Stokes wave of every stage should be sufficiently intense to serve as a pump in the next stage of the cascade. The electric fields of the previous, with respect to the fourth, cascades of SPS significantly excited vibrations of the transverse optical phonon associated with the polariton wave. Here we can talk not only about the excitation of the dipole by the Z-component of the electric fields of the pumping waves but also about the biharmonic excitation of oscillations by a pair of intense phased light waves with a frequency difference close to the natural frequency of this oscillation. By the way, the high degree of excitation of the vibration with frequency of 5.64 THz is also evidenced by radiation at a wavelength of 526.7 nm (the anti-Stokes scattering component with a pump wavelength of 532 nm and a vibration frequency of $\sim 188$ cm$^{-1}$), which can be detected near the pump beam spot. Due to the huge intensity of the pump wave (relative to the radiation at the anti-Stokes frequency) and the need to significantly weaken the signal at the entrance to the optical fiber for transportation to the spectrograph, the line on the spectrum looks weak but is confidently recorded. Thus, the gradual rocking of the oscillation as the subsequent cascades are turned on can lead to the manifestation of anharmonism of the transverse optical phonon associated with the polariton. Since the SPS cascades are separated in time [26], anharmonism is expected to manifest itself precisely in cascades (*beans*) of high orders. Due to anharmonicity, the frequency of the polariton wave decreases, which may well lead to the values given in Table 3 for the polariton of the fourth cascade. However, this is still a hypothesis, and our experimental data do not provide grounds for either refutation or proof of it.

One can see from Figure 4 that along with the cascaded SPS spectral components, the spectral lines corresponding to the scattered pump radiation (532 nm) and to the Raman scattering on longitudinal optical phonons (539.4 and 551.4 nm) are measured in different *beans*. These two last lines correspond to the Stokes shifts of $\sim 258$ and $\sim 661$ cm$^{-1}$. Raman Stokes shifts were measured in [8,31] in the case of a KTP crystal of another configuration: *X*-cut KTP crystal and polarization of the pump beam parallel to *Z*-axis. The following Raman shifts were measured: $\sim 266$ and $\sim 694$ cm$^{-1}$. In our case, the pump beam propagated in the *XOZ* crystal plane but not along the *X*-axis, and its polarization was parallel to this plane. The difference in Raman shifts between our work and the experiments performed in [31,32] may be due to the different configurations of the KTP crystals.

## 5. Conclusions

Cascaded stimulated polariton scattering (SPS) was investigated in a KTP crystal under pumping by powerful picosecond pulses of a 532 nm wavelength. In our work, polariton components are organized as regular sequences with increasing angles between each next signal component and the pump beam. Five polariton components were detected above the pump beam with an angle of ~2° between. The difference in frequencies between neighboring polariton components was ~188 cm$^{-1}$ (~5.64 THz), with a permanently increasing angle between each next component and the pump beam.

The connection found between the polarization of the pump wave and the orientation of the crystal allows switching between multiple up- and down-conversion processes and cascaded processes of stimulated Raman scattering on phonons and polaritons. The rotation of the pump polarization controls the competition of all these processes in the nonlinear crystal. Understanding the patterns of these processes will provide the key to controlling light through light.

**Author Contributions:** Investigation, K.A.V., A.K.V., V.B.M. and V.G.T. All authors contributed to the manuscript. We confirm that neither the manuscript nor any parts of its content are currently under consideration or published in another journal. All authors have read and agreed to the published version of the manuscript.

**Funding:** This research was partially supported by the Moscow State University Program of Development.

**Institutional Review Board Statement:** Not applicable.

**Informed Consent Statement:** Not applicable.

**Data Availability Statement:** Data are contained within the article. Details are available on request.

**Acknowledgments:** The authors thank G.H. Kitaeva for fruitful discussions of the polariton processes.

**Conflicts of Interest:** The authors declare no conflict of interest.

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
