# Peer review of "Cascaded Stimulated Polariton Scattering in a Single-Pass KTP Crystal under Picosecond Pumping"

_photonics, doi:10.3390/photonics10121355_

Round 1

Reviewer 1 Report

Comments and Suggestions for Authors

This paper investigates the cascaded stimulated polariton scattering in KTP crystal pumped by powerful 63 ps pulses of 532 nm wavelength. The experimental observation is performed first, and the theoretical analysis and discussion are given later. The paper is well written. It is helpful for the people in the laser and nonlinear optics fields. I think it can be published in Photonics after minor revision.

1.      Please pay attention to the significant digits of the experimental data. For example, 553.9±0.05 should be 553.90±0.05, 532±0.02 should be 532.00±0.02, 187.84±0.7 should be 187.8±0.7, 187.86±1 should be 187.9±1.0.

2.      1064,2 nm should be 1064.2 nm.

3.      In the Introduction, Ref. [4] is said to be about the nonlinear crystal of KNbO3, actually it is not. Please check.

Author Response

Dear Reviewer,

Thank you very much for reviewing our manuscript and for your comments.

Our response is in attached file.

Sincerely,

Authors

Reviewer 2 Report

Comments and Suggestions for Authors

In my opinion this paper is not suitable for publication in the present form:

-       The authors mention in the Introduction section that polariton scattering has been studied from the first experimental work in 1970. However, it was not mentioned or remarked the applications or possible applications that this kind of system have or could have currently.

- It is necessary to include more updated bibliography 

- Figure 4 must be improved, the spectra must be show complete.

- FIgure 5 and Figure 6 are not cited in the text.

- In Figure 5, please review the figure caption and the color lines mentioned.

Author Response

(The authors gave the same response as above.)

Reviewer 3 Report

Comments and Suggestions for Authors

The article under review is well and logically written. Just some small questions:
-why lenses L1 and L2 located in front of mirror M2 and not after it?
-probably in Figure 7, the wavevectors for different polaritons should be of different colors?

Author Response

(The authors gave the same response as above.)

Reviewer 4 Report

Comments and Suggestions for Authors

1-     In Abstract:

·        What is the main novelty of this research? It is not clear from abstract or at the end of the introduction part. For that, the introduction part must be included the different in this work in compared to the previous similar work.

2-    In introduction

·       The introduction need to be supported with similar literature survey to show the different in this work in compared to others

3-     In the preparation method:

·        The name and model of used picosecond laser device must be inserted?

·        It was recommended to enhance the quality of the schematic diagram to a colored form to clarify the changes of the laser wavelength.

·        Specification of CCD-detector must be inserted

4-     Language,

·        The subscript and superscript must be carried in the whole manuscript as (LiNbO3).

·        Also, check the typing of each fig. as it sometimes fig.3 and sometimes fig. 4.

·        The punctuation between keywords must be “;”

·        In figure caption 4: it must be (a), (b), (c), and (d). So, The language need to be checked to increase the quality of the paper

5-     In conclusion

·        The authors should refrain from summarizing the work in the conclusion; instead, only the main findings should be placed.

6-     In references

·        The references must be supported with more recent articles as the recent used reference is 2020 

Comments on the Quality of English Language

it to be enhanced

Author Response

(The authors gave the same response as above.)

Round 2

Reviewer 2 Report

Comments and Suggestions for Authors

The authors have take into account the comments.

Reviewer 4 Report

Comments and Suggestions for Authors

it is accpeted in the current form

Comments on the Quality of English Language

it is accepted